# Inhaled Corticosteroids in Subjects with Chronic Obstructive Pulmonary Disease: An Old, Unfinished History

**DOI:** 10.3390/biom14020195

**Published:** 2024-02-06

**Authors:** Andrea S. Melani, Sara Croce, Gaia Fabbri, Maddalena Messina, Elena Bargagli

**Affiliations:** Respiratory Diseases Unit, Department of Medicine, Surgery and Neurosciences, University of Siena, 53100 Siena, Italy; sara.croce@hotmail.it (S.C.); gaiafabbri4@gmail.com (G.F.); messina18@student.unisi.it (M.M.); bargagli2@gmail.com (E.B.)

**Keywords:** chronic obstructive pulmonary disease, inhaled corticosteroid, exacerbations, pneumonia, safety, single-inhaler triple therapy, quality of life, lung function, symptoms, survival

## Abstract

Chronic obstructive pulmonary disease (COPD) is one of the major causes of disability and death. Maintenance use of inhaled bronchodilator(s) is the cornerstone of COPD pharmacological therapy, but inhaled corticosteroids (ICSs) are also commonly used. This narrative paper reviews the role of ICSs as maintenance treatment in combination with bronchodilators, usually in a single inhaler, in stable COPD subjects. The guidelines strongly recommend the addition of an ICS in COPD subjects with a history of concomitant asthma or as a step-up on the top of dual bronchodilators in the presence of hospitalization for exacerbation or at least two moderate exacerbations per year plus high blood eosinophil counts (≥300/mcl). This indication would only involve some COPD subjects. In contrast, in real life, triple inhaled therapy is largely used in COPD, independently of symptoms and in the presence of exacerbations. We will discuss the results of recent randomized controlled trials that found reduced all-cause mortality with triple inhaled therapy compared with dual inhaled long-acting bronchodilator therapy. ICS use is frequently associated with common local adverse events, such as dysphonia, oral candidiasis, and increased risk of pneumonia. Other side effects, such as systemic toxicity and unfavorable changes in the lung microbiome, are suspected mainly at higher doses of ICS in elderly COPD subjects with comorbidities, even if not fully demonstrated. We conclude that, contrary to real life, the use of ICS should be carefully evaluated in stable COPD patients.

## 1. Introduction

Chronic obstructive pulmonary disease (COPD) is a heterogenous syndrome with many different phenotypes, characterized by persistent respiratory symptoms and incompletely reversible airflow obstruction. COPD is one of the major causes of disability and death. Although COPD is a chronic condition, it is often punctuated by acute events, defined exacerbations that negatively impact on lung function, quality of life, and survival [1].

Regular maintenance use of inhaled bronchodilator Long-Acting Beta2-Agonist (LABA) and Long-Acting Muscarinic receptor Antagonist (LAMA) is the cornerstone of COPD pharmacological therapy [1]. These drugs are available with separate inhalers or in a single inhaler containing two drugs. Inhaled corticosteroids (ICSs) are another class of inhaled drugs known for their anti-inflammatory properties. ICSs are the cornerstone for the treatment of asthma, the other largely diffused chronic obstructive airway disease, but are also utilized in COPD. The use of ICS in COPD subjects has had ups and downs, and the final word on its therapeutic significance is still ongoing.

This narrative review evaluates the indications, efficacy, and safety of long-term regular ICS use in subjects with COPD; we also investigate the differences between various molecules of ICS, if any. A systematic search was conducted on Medline through PubMed, including studies published up to October 2023. For the study search, we used the following words: (“Pulmonary Disease, Chronic Obstructive” OR “COPD” OR “Chronic obstructive pulmonary disease” OR “Pulmonary Disease, Chronic Obstructive”, OR “Emphysema” OR “Chronic bronchitis”) AND (“inhaled corticosteroids”, beclomethasone”, “budesonide”, “triamcinolone”, “fluticasone propionate”, “fluticasone furoate”, “mometasone”, “ciclesonide”, “flunisolide”. We included narrative and systematic reviews, randomized controlled trials (RCTs), and observational studies focused on adult COPD subjects.

## 2. History of ICS Use in COPD Subjects

From the early 2000s, studies investigating the significance of ICSs alone in COPD subjects found statistically significant improvements in symptoms and quality of life (as measured by the St George’s Respiratory Questionnaire, Mean Difference −1.22 units/year, 95% Confidence Interval, CI −1.83–−0.60), a reduction in exacerbation rate (Rate Ratio, RR 0.88 exacerbations per participant per year, 95% CI 0.82–0.94) and, less evident, a decline in forced expiratory volume in one second (FEV1, the most widely used parameter of lung function in COPD subjects; mean difference 6.31 mL/year benefit, 95% CI 1.76–10.85), but no difference in mortality rate (odds ratio, OR 0.94, 95% CI 0.84–1.07) vs. placebo [2]. The outcome of reduced mortality is very important, but difficult to achieve for the relatively small number of events, requiring large populations and long follow-up duration. Likewise, the combination of ICS and LABA (ICS/LABA) was more effective in reducing exacerbations than ICS alone (RR 0.87, 95% CI 0.80–0.94) [3]. These results supported the diffusion of ICS/LABA combinations in real life. In 2016, the RCT eFfect of indacaterol/glycopyrronium vs. fluticasone/salmeterol on COPD Exacerbations (FLAME) study, that enrolled subjects with a relatively low risk of exacerbation (a condition that more strictly reflects real life: only 20% of participants had two or more episodes per year), found that a combination of LABA and LAMA (LABA/LAMA) was more effective to prevent exacerbations (RR 0.89; 95% CI 0.83–0.96; *p* = 0.003) than an ICS/LABA in COPD patients. This result raised some doubts on the role of the ICS/LABA as the first choice for the regular treatment of stable COPD subjects [4]. However, rather than removing the ICS, physicians started to prescribe both LABA/ICS and LAMA inhalers to treat COPD subjects. Until recently, triple therapy with ICS, LABA, and LAMA was delivered with multiple inhalers (MITT, multiple inhaler triple therapy). Single-inhaler triple therapy (SITT) has been a more recent development for the treatment of COPD. SITT has the potential of a simpler dosing regimen than MITTs, thus hopefully improving adherence. The INvestigation of TRelegy Effectiveness: usual PractIce Design (INTREPID) study showed that the use of SITT improved health status, as evaluated by the COPD Assessment Test (CAT), more than the MITT [5]. Other retrospective or prospective studies including pharmacy data suggest that the use of SITT may improve adherence, persistence, and, sometimes, symptoms and exacerbations compared with MITT [6,7,8].

Two recent RCTs evaluating the value of SITT in COPD subjects found improved survival compared with LABA/LAMA users. The InforMing the PAthway of COPD Treatment (IMPACT) trial [9] compared SITT, LABA/LAMA, and ICS/LABA using the same ICS (fluticasone furoate 100 mcg), LABA (vilanterol), and LAMA (umeclidinium) and the same device, the Dry Powder Inhaler (DPI) Ellipta. The Efficacy and safety of triple THerapy in Obstructive lung diSease (ETHOS) trial [10] compared SITT, LABA/LAMA, and ICS/LABA using the same ICS (budesonide at 200 and 400 mcg twice daily for the SITT arm, but only at the lower strength for the ICS/LABA arm), LABA (formoterol), and LAMA (glycopyrrolate) and the same Metered Dose Inhaler (MDI) device. The IMPACT [9] and the ETHOS [11] trials reported that the use of SITT reduced all-cause mortality over one year compared with the LABA/LAMA (for the IMPACT: Hazard Ratio, HR 0.72; 95% CI 0.53–0.99; *p* = 0.042; for the ETHOS, HR 0.51; 95% CI, 0.33–0.80; *p* = 0.0035). Previous studies, such as the TOwards a Revolution in COPD Health (TORCH) (HR 0.825; 95% CI, 0.68–1.00; *p* = 0.052) [12] and the Study to Understand Mortality and Morbidity In COPD Treatment (SUMMIT) (HR 0.88; 95% CI 0.74–1.04; *p* = 0.137) [13] failed to find a reduced mortality (primary outcome of the studies) in COPD patients using ICS/LABA compared with a placebo. The results of the IMPACT and ETHOS trials have been criticized [14]. Firstly, differently from the TORCH and the SUMMIT, these studies selected subjects with a history of common exacerbations; second, mortality was not the primary outcome for both studies; third, subjects with a current but not previous history of asthma were excluded; fourth, the study design required stopping the ICS during the run-in (approximately 70% and 80% of subjects enrolled in the IMPACT and the ETHOS studies, respectively, were using ICS before the study entry), and the excess of mortality in non-SITT users was only limited to the first 90 days of active treatment [14]. The survival gains obtainable with the SITTs compared with dual bronchodilation observed in the IMPACT and the ETHOS trials seem to be mainly related to the greater reduction in COPD exacerbations. The IMPACT trial found that the SITT assured a lower moderate-to-severe exacerbation rate (primary outcome of the study) than the LABA/LAMA (RR 0.75; 95% CI, 0.70–0.81; *p* < 0.001) or the ICS/LABA (RR 0.85; 95% CI, 0.80–0.90; *p* < 0.001) [9]. The ETHOS trial showed that the SITT ensured a lower moderate-to-severe exacerbation rate than the LABA/LAMA (RR 0.76; 95% CI, 0.69–0.83; *p* < 0.001 for the high ICS strength; RR 0.75; 95% CI, 0.69–0.83; *p* < 0.001 for the low ICS dosage) and the LABA/ICS (RR 0.87; 95% CI, 0.79–0.95; *p* < 0.003 for the high ICS strength; RR 0.86; 95% CI, 0.79–0.95; *p* = 0.002 for the low ICS dosage arms). [10]. Likewise, exacerbations are very important events, as one single moderate-to-severe episode may yield a decline in lung function [15] and all-cause mortality [16]. A pooled analysis of the other currently available SITTs containing formoterol as LABA, glycopyrronium as LAMA, and beclomethasone dipropionate (BDP) as ICS via the MDI in COPD patients also found significant reductions in moderate-to-severe exacerbations, but not in all-cause mortality rate (HR 0.71, 95% CI 0.50–1.02; *p* = 0.066) as compared with the LABA/LAMA [17]. A pooled analysis of data from six RCTs, including over 6000 COPD subjects with low risk of exacerbation (about 81% of participants with 0–1 exacerbation in prior year), did show no differences in mortality rate (HR 1.06; 95% CI 0.68, 1.64; *p* = 0.806) between triple therapy (MITT, not SITT) and LABA/LAMA [18]. 

## 3. Biomarkers Useful to Predict Response to ICS in COPD Patients

Not all COPD subjects with a high risk of moderate-to-severe exacerbations are responsive to the use of ICS. This is not surprising, as COPD is a heterogeneous condition, constituting several endotypes and phenotypes with differences in histology, clinics, chest imaging, lung function, comorbidities, and disease progression [19]. It is of paramount importance to have easy-to-use biomarkers that can objectively and reliably distinguish between ICS responders and non-responders. COPD is characterized by prevalent neutrophilic inflammation, which has poor responsiveness to ICS, but a subset (20–30% of total cases) show an eosinophilic phenotype [19]. Several post hoc and retrospective analyses of large RCTs have shown that blood eosinophil counts may help to predict the response of ICS added to dual bronchodilator treatment in reducing the risk of exacerbations. Higher blood eosinophil counts (≥300/mcl) were associated with good responses to ICS, while no or small results had been found by adding an ICS in stable COPD subjects with eosinophil counts lower than 100 cells/µL. The thresholds of <100 cells/µL and ≥300 cells/µL have been suggested to predict different probabilities of treatment benefit, but they should be considered as estimates, rather than precise cut-off values [20]. Blood eosinophil counts seem to be relatively stable over time in a large primary care population and scarcely influenced by prior ICS use, although considerable overlaps have been observed at higher levels (≥300 cells/µL) [20]. Other factors may influence the significance of blood eosinophil count in predicting the risk of exacerbation. Possibly, tobacco smoking is the most widely known factor that can negatively influence the responsiveness to ICS in COPD subjects with common exacerbations [21]. This negative interaction is mainly due to a reduction in histone deacetylase-2 enzyme activity [21].

Some studies have investigated whether COPD subjects with high blood eosinophil counts that do not receive ICS show greater declines in lung function. A post hoc analysis of the RCT Inhaled Steroids in Obstructive Lung Disease in Europe (ISOLDE) trial [22], where COPD subjects with moderate-to-severe airflow obstruction were randomly treated with ICS alone or placebo during a 3-year follow-up, did show no differences between treatments, but the subgroup using ICS and blood eosinophil counts > 2% showed a reduced post-bronchodilator decline in FEV1 over time. Another study found that COPD patients with high blood eosinophil counts (>350 cells/L) and at least one exacerbation had a significantly greater FEV1 decline if they were not treated with ICS [23]. In addition, COPD subjects with a baseline blood eosinophil count ≥300 cells/µL had a greater FEV1 loss [24] after withdrawal of ICS.

Researchers have searched for other biomarkers able to predict the response of ICS in COPD subjects. The Groningen Leiden Universities Corticosteroids in Obstructive Lung Disease (GLUCOLD) study, performed with a cohort of 114 COPD subjects, found that normal or almost normal lung diffusion capacity, a high degree of bronchial responsiveness, imaging findings of bronchiolitis, and low air trapping were predictors of response to ICSs [25]. These findings are very interesting and useful to select some ICS-sensitive phenotypes, but there is a need for more precision to personalize the prescription of ICS on an individual basis in COPD subjects. Pharmacogenomics might possibly help to distinguish endotypes that benefit from ICS use [26]. Molecular analyses of bronchial biopsies collected in the GLUCOLD study found increased T2-related gene expression in the subset of COPD subjects with high blood eosinophil counts and good responses to ICS [27]. Genetic studies could also identify subjects associated with an increased decline in lung function, but responsive to ICS [28] or subgroups with high risk of side effects to ICS, such as susceptibility to pneumonia [29] or to adrenal failure [30].

## 4. ICS: Indication

The Global initiative for chronic Obstructive Lung Disease (GOLD) COPD guidelines [1] do not recommend use of ICS alone in stable COPD subjects, but in association with bronchodilators. GOLD reports [1] strongly support the use of SITT in COPD subjects with a history of asthma or as step-up pharmacologic therapy in the presence of moderate-to-severe exacerbations plus blood eosinophil count ≥ 300/mcl, but weakly favor the use of SITT if blood eosinophil count ≥ 100 cells/mcl [1]. The American Thoracic Society clinical practical guidelines emphasize that most studies showing benefits of SITT on survival have selected patients with common exacerbations, while benefits of SITT are less clear in COPD subjects who had experienced zero to one exacerbation in the past year [31]. An official European Respiratory Society document also suggests that ICS withdrawal is feasible in COPD patients without a history of concomitant asthma or frequent exacerbations, and step-down should be considered in the absence of improvement in baseline symptoms [32]. A prospective 3-year study showed that slightly more than 50% of COPD subjects did not show any exacerbation during the study period, and 23% had at least two episodes per year [33]. Other studies have found that approximately one-third of COPD subjects may be estimated as common exacerbators [34,35]. However, the indication of a product license for currently available SITTs is slightly larger and includes all patients who remain uncontrolled despite LABA/LAMA or ICS/LABA maintenance therapy, including subjects with persistent symptoms and not only those with recurrent exacerbations. In addition, neither the Food and Drug Administration (FDA) nor the European Medicines Agency (EMA) recommend the use of blood eosinophil counts for the choice of pharmacological treatment in COPD subjects.

Many asthmatics acquire clinical characteristics of COPD over time. This co-existence is common and has been defined as Asthma COPD Overlap Syndrome (ACOS). Studies estimate that up to 15–25% of subjects diagnosed with COPD are suspected to have concomitant asthma [36,37]. Subjects with ACOS should receive a prescription of combinations containing ICS [1]. However, it is not always easy to accurately identify subjects with ACOS, as diagnosis is based on rather vague characteristics, such as a history of asthma, respiratory symptoms before the age of 40 years, the presence of atopy and allergies, and high responsiveness to bronchodilators. Hopefully, pharmacogenomics might help the clinician to individuate the asthma–COPD overlap phenotype, that, at present, often remains poorly defined [26].

## 5. ICS: Safety

### 5.1. Generalities

The use of ICS is not free from adverse events. These effects may be distinguished into local and systemic adverse events. Local side effects are due to corticosteroid particles settling into the mouth, throat, larynx, and lung, although the ultimate cause has not been fully defined. Some prevention measures, such as rinsing the mouth with water immediately after inhalation and/or using a valved holding chamber (for MDIs), can help minimize them.

Systemic side effects are related to the drugs that enter the bloodstream either through the gastrointestinal route, from a drug that is swallowed, or through lung absorption, from a drug released into the lower airways.

### 5.2. Mouth and Upper Airways Disturbances, Pneumonia, Other Lung Infections, and Changes in the Lung Microbiome

Mouth and upper airway side effects are very common when using ICS, although are usually not so severe as to sustain the discontinuation of treatment [2]. The use of ICS is associated with an almost three-fold higher risk of oral candidiasis (OR 2.66, 95% CI 1.91–3.68) [2] compared with non-users. It is proposed that the inhibition of normal host defense functions at the oral mucosal surface can lead to the occurrence of oral candidiasis. All ICSs may yield oral candidiasis, but the use of fluticasone propionate use seems to involve greater risk than budesonide or BDP [38]. The use of ICS is associated with a 3–5-fold higher risk of dysphonia vs. non-ICS users [38,39]. Hoarseness is also common (OR 1.98, 95% CI 1.44–2.74) in ICS users [2], is dose related, and, possibly, is due to steroid myopathy of the laryngeal muscles [39].

It has been observed that ICS use in asthmatics is associated with an increased risk of obstructive sleep apnea syndrome, a common disease characterized by repeated pharyngeal collapse during sleep. Local deposition of ICS might induce muscle fiber atrophy and central neck fat redistribution, predisposing to apnea–hypopnea episodes [40].

The use of ICS in COPD subjects increases the risk (OR 1.38, 95% CI 1.02, 1.88) of pneumonia [41]. An extensive review [42] evaluating the side effects of ICS in COPD subjects updated to October 2020 confirmed that long-term exposure (≥1 year) to ICSs increased the risk of pneumonia by 41% based on 19 RCTs (RR 1.41, 95% CI 1.23–1.61; *p* < 0.00001) and by 26% based on 21 cohort studies (HR 1.26, 95% CI 1.14–1.38, *p* = 0.0001). A relationship exists between the risk of pneumonia and the dosage of ICS [42,43]. In a nested case–control study, withdrawal of ICS was associated with a 37% decrease in the rate of pneumonia [44]. In addition, the risk of pneumonia might be linked to the used ICS, the highest probability being associated with fluticasone propionate and fluticasone furoate [42]. ICSs that remain for a longer time into the lower airways might induce greater local immunosuppression, predisposing to pneumonia [42]. However, studies on ICSs have different criteria for recording pneumonia, and it is difficult to draw firm conclusions in the absence of head-to-head comparison trials. The EMA’s Pharmacovigilance Risk Assessment Committee (PRAC) recognized pneumonia as a class effect of ICS-containing therapies in patients with COPD, with no conclusive evidence of intra-class differences [45]. Patients who currently smoke, with a body mass index < 25 kg/m^2^, severe airflow obstruction, [46], chronic bronchial infection or associated bronchiectasis [47], and low blood eosinophil counts (<2%) [48] have a greater risk of developing pneumonia. Although pneumonia is a common cause of sepsis, the risk of sepsis does not seem to be increased using ICS, even at high doses [49]. Some cohort studies found that the use of ICSs increased the risk of developing tuberculosis, especially at higher doses [50]. ICSs also predispose people to an increased risk of non-tuberculous mycobacteria infections (OR 4.46 95% CI 2.13–9.95) [51].

It is increasingly recognized that COPD progression and exacerbation frequency, even in the early phases, are linked to the composition of the lung microbiome [52,53]. In addition, COPD subjects can modify the lung microbiome and acquire neutrophilic or eosinophilic inflammatory predominant patterns during exacerbations [52,53,54,55,56]. The lung microbiome may also influence the sensitivity to ICS. The predominance of proteobacteria, such as Hemophilus influenzae, sustains the production of Interleukin-(IL)8 and neutrophil recruitment, which is usually associated with poor responsiveness to ICS [56]. Likewise, the risk of overgrowth with bacteria associated with IL-8 production and neutrophil recruitment seems to be more common in ICS-treated COPD patients with low blood eosinophil counts [56]. Long-term use of ICS itself might contribute to potentially deleterious airway microbiota changes by increasing the bacterial load of potentially pathogenic germs. ICS use in COPD subjects has been associated with dose-dependent increased risks of acquiring *Haemophilus influenzae* [57], *Streptococcus pneumoniae* [58], *Moraxella catarrhalis* [59], and *Pseudomonas aeruginosa* colonization [60]. Leitao et al. evaluated the effects of a 12-week treatment with ICS/LABA combination vs. LABA alone on the airway microbiome of stable COPD subjects via bronchoalveolar lavage. They found a reduced heterogeneity in the lung microbiome for the ICS/LABA-treated group compared with the LABA-only arm [61]. ICS use might differently inhibit phagocytosis between germs, leading to decrease in bacterial diversity and increased abundance of some bacteria in COPD subjects. Scott et al. [62] showed that phagocytosis was impaired by fluticasone propionate, beclomethasone, and budesonide in a dose-dependent manner, with fluticasone having the greatest inhibitory effect.

### 5.3. Systemic Side Effects

Although occasional, some individuals may experience allergic reactions after ICS use with skin rash, itching, swelling, severe dizziness, or difficulty breathing.

Long-term use of systemic corticosteroids is not recommended in COPD, because it is associated with important systemic adverse events [63]. Although inhalation reduces drug bioavailability, improving the therapeutic ratio, it is unclear whether the long-term use of ICS may yield systemic side effects, as some observational studies would seem to suggest [42]. Available data from RCTs do not show systemic effects of ICS at recommended doses in COPD subjects, but most of them have a duration of 1 year at best; this time may not be long enough for showing some adverse events. In addition, RCTs seldom enroll frail old COPD subjects with comorbidities that are particularly prone to systemic consequences of ICS treatment [63].

There was no clear evidence of increased risk for eye disorders in COPD subjects using ICSs [42], but a recent meta-analysis has suggested that ICS use at high daily doses may be associated with an increased risk of developing cataracts [64]. Some case–control studies observed an association between ICS doses and an increased risk of diabetes-related complications or deterioration [65,66,67]. Moreover, some cohort studies have shown a relationship between long-term regular use of ICS at high daily doses and a significant decline in bone mass [43], but it is unclear whether this can translate into an increased risk of fractures. A review updated to September 2022 including 61,380 COPD participants from 26 RCTs showed that exposure to ICSs did not increase the risk of fracture (RR 1.10; 95% CI 0.98–1.23, *p* = 0.10) or osteoporosis (RR 0.93; 95% CI 0.49–1.79; *p* = 0.84) [68]. However, a retrospective study on a large Swedish COPD population found a relationship between ICS use and any osteoporosis-related event. In addition, the risk of all bone fractures and fractures related to osteoporosis in COPD subjects using ICSs was higher compared with non-ICS users, with a clear dose–effect relationship [69]. In another large cohort of patients, the utilization of any ICS was not associated with an increased risk of fracture (RR, 1.00; 95% CI, 0.97–1.03), but this rate significantly increased when ICSs were used for more than 4 years and at higher dosages (RR, 1.10; 95% CI, 1.02–1.19) [70].

Although less common compared with oral corticosteroids, some people using ICSs may experience thinning of the skin and increased susceptibility to bruising, especially in elderly patients receiving high-dose ICS [71].

Homeostasis of the human body requires the presence of corticosteroid hormones that are released by the adrenal gland. The amount of delivery is strictly regulated by the hypothalamic–pituitary–adrenal (HPA) axis, with internal feed-back mechanisms. The body does not distinguish the source of released corticosteroids; thus, the prolonged administration of exogenous corticosteroid drugs may suppress the endogenous delivery of cortisol (the main corticosteroid hormone) and its abrupt withdrawal can lead to adrenal insufficiency. It is not fully defined whether long-term ICS use in COPD subjects can yield adrenal failure. One review found that adrenal failure was extremely uncommon when ICS were used under approved doses [72]. However, another seminal revision estimated adrenal insufficiency rates in about 8% of subjects undergoing withdrawal after long-term ICS use. In addition, ICS treatment longer than one year was associated with adrenal failure in about 20% of treated subjects, and there was a relationship with daily dosage [73]. More recently, although conducted in severe asthmatics, Lobato et al. reported low levels of free plasma cortisol (a surrogate measurement of adrenal failure) in a consistent proportion of subjects treated with 200 μg/day fluticasone furoate (double the daily dosage approved in COPD) [74]. Maijers et al. investigated the effect of some ICSs at different dosages in corticosteroid-dependent asthma; they showed that daily doses of 1000 μg fluticasone propionate and budesonide reduced the prednisone requirement by about 5 and 2.14 mg, respectively [75]. This result seems to confirm that the use of ICS at high daily doses may give systemic exposure [75]. In dose–response studies, Dailey-Yates et al. [76] estimated the daily dose of different ICSs that would result in 20% cortisol suppression in corticosteroid-dependent asthmatics, a surrogate measurement of systemic exposure to ICSs and interference of the HPA axis. They concluded that 620 μg/day of budesonide corresponds to 289 μg of fluticasone furoate as cortisol-equivalent exposures, suggesting that the last ICS has approximately double the systemic potency to impair the HPA axis with respect to budesonide. This comparison does not consider that the efficiency in lung drug delivery of the device Turbohaler^®^ (delivering budesonide) is double that of the Ellipta^®^ device (delivering fluticasone furoate) [77]. Moreover, the short duration of treatment (one week for this arm) may have underestimated the potential of systemic exposure attributable to the ICS with higher lipophilicity, such as fluticasone furoate.

## 6. ICS Mechanism of Action, Comparison between Molecules and Proper ICS Dosage

The corticosteroid molecule acts by transiting through cell membranes and binding to the glucocorticoid receptor (GR). Most anti-inflammatory activities are mediated by the pleiotropic effects of GR. It regulates the expression of inflammatory genes encoding the synthesis of several cytokines and binds to other transcription factors, such as nuclear factor (NF)-ĸB. Corticosteroids also exert anti-inflammatory effects through non-genomic mechanisms blocking the production of mediators associated with inflammatory cells, such as the GR-dependent inhibition of the MAPK pathway. If the cellular mechanism of action for corticosteroids is known, it is not fully recognized that the primary target of ICS in COPD [19,56].

ICSs should not be used alone in COPD subjects, but in association with inhaled bronchodilators. Several inhaler combos containing ICSs are available on the market. Budesonide, BDP, fluticasone propionate and fluticasone furoate are the ICSs approved for use in COPD subjects in Western countries (see Table 1 for available combinations). BDP is a pro-drug that is fully converted to its active moiety, beclomethasone monopropionate, from esterase located into the lungs. The efficacy of fluticasone propionate and fluticasone furoate is dependent on the intact molecule, and these two molecules should be considered as distinct. ICS marketed for use in COPD exhibits a range of potency, physicochemical properties, and pharmacokinetic parameters, as displayed in Table 2 [78,79,80,81,82,83]. Notably, the effect of a certain ICS may be modulated by the concomitant use of long-acting bronchodilator(s) [84,85], even if the clinical relevance of this interaction has not yet been fully established. Lipophilicity is a key property of ICS and is associated with higher affinity for the glucocorticoid receptor and longer retention time into the lower airways. These characteristics determine greater potency. Fluticasone propionate is more potent and fluticasone furoate is much more potent than budesonide and BDP [83]. However, the greater potency has a relatively mild clinical significance if it is related to higher systemic exposure. Daley-Yates et al. compared budesonide, fluticasone propionate, and fluticasone furoate in asthmatics through a dose–response study evaluating airway hyper-responsiveness via an adenosine monophosphate challenge and HPA suppression by measuring baseline serum cortisol levels [81]. They reject the hypothesis that ICS are therapeutically equivalent, concluding that efficacy in the lung and systemic exposure are not necessarily related and attributing advantages to ICS molecules with higher lipophilicity, such as fluticasone furoate [81]. However, this result has been criticized. Lipworth et al. [82] noted that there was no evident separation in bronchoprotection between different doses of ICSs, and the respective confidence intervals were overlapping. Moreover, the test challenge was possibly biased, being organized at 12 h post dose for all ICS when only fluticasone furoate, which is administered once daily, resulted fully effective. In addition, the duration of dosing for this arm was only 1 week, which is too short a period to establish any clinically meaningful degree of HPA suppression, favoring ICSs with higher lipophilicity [82]. Other authors [83] dispute that long retention in the airways is necessarily advantageous for ICS molecules. Effectively, one study has shown that the rate of expectorated fluticasone propionate on the dose released into the lungs after the inhalation of LABA/ICS combinations in COPD subjects was significantly higher than that of budesonide [86]. Accordingly, a relatively fast dissolution time might minimize removal from the airways by mucociliary clearance and be advantageous. In addition, longer pulmonary residence times might also enforce the topical immunosuppressive action of the ICS, predisposing to increased risk of pneumonia or unfavorable changes in lung microbiome [42].

Fluticasone propionate and fluticasone furoate have other interesting peculiarities, as they show complete first-pass hepatic metabolism and no bioavailability of the drug dose swallowed with respect to budesonide and BPD [78,79,80]. This property is particularly important because the Diskus^®^ and Ellipta^®^ inhalers, releasing fluticasone propionate and furoate, respectively, have lower drug lung delivery than those delivering budesonide and BDP [77]. These latter devices produce finer aerosols (said extra-fine aerosols) that have been associated with improved inhaler techniques and reduced risks of pneumonia hospitalization [87,88]. However, apart from greater drug lung deposition, it is not fully known if extra-fine aerosols can effectively assure improved outcomes in COPD.

Head-to-head trials comparing SITTs containing different ICSs have not yet been performed. A systematic review found that there was no substantial difference between available SITTs regarding reductions in exacerbation rates and the use of rescue medications, improvements in lung function, quality of life, and symptoms [89]. In contrast, another review suggests that once-daily SITT might provide superior improvements in lung function than other available SITTs [90], but further confirmation is required.

Another point requiring other investigation is the optimal dosage of ICSs in COPD subjects. In Europe, the recommended dosage of ICSs in COPD subjects using the ICS/LABA fluticasone propionate/salmeterol is 500 mcg twice daily, while in US the approved dosage of fluticasone propionate is 250 mcg twice daily. The same manufacturer proposed RCTs and received indications for the treatment of COPD with the SITT delivering fluticasone furoate at a dosage of 100 mcg daily, which should correspond to fluticasone propionate 250 mcg twice daily. Few studies have compared different ICSs and a range of doses comparing both efficacy and safety endpoints in COPD subjects. In the ETHOS study, a reduction in mortality was only observed with the SITT containing budesonide at a dose of 400 mcg, but not at 200 mcg twice daily [8]. However, both ICS dosages assured the same reduction in exacerbation rate (a key driver in achieving improved survival) with respect to the LABA/LAMA. One review [91] including 13 RCTs did not observe any difference in moderate-to-severe exacerbation (RR: 1.01, 95% CI: 0.96–1.06) or pneumonia risk (RR: 1.07, 95% CI: 0.86–1.33, I2: 9.3%) between high (800 mcg budesonide or equivalent) and low-medium (400 mcg budesonide) ICS doses. A recent meta-analysis including 60 RCTs confirmed that triple inhaled therapy containing ICSs (OR, 0.73; 95% CI, 0.59–0.91) was associated with a reduction in the all-cause mortality risk among COPD; subgroup analyses revealed that both medium (OR, 0.71; 95% CI, 0.56–0.91) and low ICS doses (OR, 0.88; 95% CI, 0.79–0.97) were involved in this association [92]. However, the same concept of low and high ICS dosages, usually drawn from asthma, is not well defined in COPD, even if it is evident that the findings in asthma cannot be translated into COPD [19].

## 7. Conclusions

ICSs alone should not be used in stable COPD subjects, but in association with inhaled bronchodilators The GOLD COPD 2023 report [1] strongly recommends the use of SITT in COPD subjects with a history of asthma or as step-up pharmacologic therapy on top of dual bronchodilators in the presence of hospitalization for exacerbation or at least two moderate exacerbations in the last year [1]. A high blood eosinophil count (≥300/mcl) is an adjunctive parameter useful to define the subset of COPD subjects responsive to ICS. SITT is contraindicated in COPD subjects with a history of repeated pneumonia, mycobacterial infections, blood eosinophil count < 100/mcl [1]. It has been estimated that approximately one-third of COPD subjects should benefit from ICS [34,35,36,93]. In contrast, in real life, the use of MITT and, more recently, SITT, is increasing as starting treatment, independently of clinics and the presence of exacerbations [93,94,95]. In addition, step-down is very difficult to observe [96]. Possibly, physicians feel that COPD subjects who remain poorly controlled despite dual bronchodilators should receive triple inhaled therapy in the hope of relieving symptoms, preventing exacerbations and disease deterioration. The after-thought of this attitude is that benefits of ICS use outweigh risks. Some findings can support these beliefs. In the KRONOS RCT [97], benefits of SITT on several outcomes were observed in symptomatic COPD patients without a history of exacerbations. In addition, slightly more than one-third of COPD patients with no exacerbation at baseline will experience an episode within the subsequent three years [98]. There is a need for RCTs to determine the role of SITT in symptomatic COPD subjects with occasional exacerbations and mild-to-moderate airflow obstruction [99]. As RCTs enroll highly selected populations and are not well generalizable, observational studies should bridge the gap between RCTs and real life and clear the consequences of ICS use for long-term period and at higher doses, if any, on lung microbiome and systemic exposure in elderly COPD subjects with multiple comorbidities. Other studies should clarify whether the anti-inflammatory effect and the systemic exposure of different ICSs work hand in hand, so that these molecules may be considered as interchangeable after adjustment for differences in potency. Research on the optimal dosage of ICS in COPD subjects would also be welcomed.

We conclude that the role of ICSs in COPD subjects is not ultimately defined.

## Figures and Tables

**Table 1 biomolecules-14-00195-t001:** Available ICSs as combinations for COPD use.

Medication Class	Agents	Inhaler Device
ICS/LABA	Fluticasone propionate/Salmeterol	DPI/MDI
	Budesonide/Formoterol	DPI/MDI
	BDP/Formoterol “	DPI/MDI
	Fluticasone furoate/Vilanterol	DPI
ICS/LABA/LAMA	BDP/Formoterol/Glycopyrronium “	DPI/MDI
	Fluticasone furoate/umeclidinium/vilanterol	DPI
	Budesonide/Formoterol/	MDI
	Glycopyrronium	MDI
	Cicloesonide/formoterol/tiotropium ^	MDI

Abbreviations: ICS, inhaled corticosteroid; LABA, long-acting β2-agonist; LAMA, long-acting muscarinic antagonist; MDI, Metered Dose Inhaler; DPI, Dry Powder Inhaler. ^ marketed in India; “ not marketed in the US.

**Table 2 biomolecules-14-00195-t002:** Some characteristics of currently marketed inhaled corticosteroids for COPD subjects.

ICS	GRC Binding Affinity	Half- Life, h	Water Solubility, μg/mL “	Clearance, h	Volume of Distribution at Steady State, L	Protein Binding, %	F, % os	Equipotent Dose
Budesonide	935	2.8	16	84	180	91	10	100
Fluticasone propionate	1775	>14	0.14	69	318	99	1	50
BDP	53	0.1	0.13	120	20	92	11	100/50 ^
BMP *	1345	2.7	15.5	120	424	92	NA	NA
Cicloesonide	12	0.4	<0.1	228	300	99	1	40
Des-cicloesonide	1200	4	7	350	1100	99	NA	NA

Legend: ICS, inhaled corticosteroid; BDP, beclomethasone dipropionate; BMP, Belomethasone mopnopropionate; * BMP is the active metabolite of BDP; DES-cicloesonide is the active metabolite of ciclesonide. Glucocorticoid receptor binding affinity is evaluated with reference to a receptor affinity of dexamethasone of 100; F, bioavailability of the swallowed dose determined in healthy subjects; ^ For HFA Modulite formulation; “ at 37 °C; NA, not available.

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
