# Peer review of "Inhaled Corticosteroids in Subjects with Chronic Obstructive Pulmonary Disease: An Old, Unfinished History"

_biomolecules, 2024, doi:10.3390/biom14020195_

Round 1

Reviewer 1 Report (Previous Reviewer 1)

Comments and Suggestions for Authors

Dear editors and authors, the article presented to me for review has been significantly improved. Below, please find a few remarks on the text. 

Line 42 - chrocni - change to chronic

Line 93 - lack of space

Line 98 - lack od space

Line 120 - lack of bracket: [5

Line 381 - lack of "r" in  prior

Line 656 - double s - wass

Line 1254 - remove 2. ICS: indication

Line 1281 - lack of space

Line 1300 - should be 4. ICS safety instead of 5. ICS safety

Line 1421 -  [48 4] - change

Linw 2154 - lack of space

Line 2644 - change: clinic al, the same for interact ion

In table: Cicloesonide/formoterol/tiotropiu   - lack of m

Line 3264 - numbers???

Author Response

We would like to thank the referee for the comments. We uploaded a revised manuscript and a revised online supplement file (with tracked changes in both files). We have modified all the points as the referee's suggested

Reviewer 2 Report (Previous Reviewer 2)

Comments and Suggestions for Authors

This manuscript provides a review of the use of inhaled corticosteroids in COPD.  The paper is well written and covers the major clinical trials that studied sufficient numbers of subjects to make the results meaningful. The ICS mechanisms section is a nice addition to a review, and helps to explain some of the differences observed in the trials.  Table 2 is also helpful for clinical practice, interpreting study results and trial design.  

General comments:

- There are over 112 references published using the search ICS and COPD.  Since most of the references cited are older that 2022, I would recommend the authors check for updates.  For example, Chen et al. ( Chest 2023) reviews the impact of ICS on mortality (no I am not one of the authors).

- For the reader who has a plethora of reviews on the subject to choose from, it might be helpful to state up front why this review is important e.g. new data.  The abstract states that a review of mortality will be discussed, but there is more to this review.

-I prefer to have the acronym of the study spelled out the first time it is used.

- It would be helpful to have a section following the review of the trials in the introduction (or somewhere) listing the trial design weaknesses.  For example most are 52 weeks or less, do they account for endotype/phenotype differences in COPD.

- Line 64 should have a reference added.

- There are several references in the literature about % of non-responders to ICS in COPD that would be a nice addition

- Do steroids inhibit neutrophils in COPD?

-The sections on steroid side effects is well done with respect to risk-benefit.  Perhaps some discussion on reduction of oral steroids by ICS would be of interest.

- The effect of patient compliance on utilizing SITTs vs MITTs vs tailoring doses would be an interesting addition.

Minor:

-Check that track changes did not add/eliminate spaces.

- line 51- backstone of cornerstone?  line 32 We ICS?  etc

-line 209-"of hospitalization"

-line 164 needs reference

Author Response

We would like to thank the referee for the comments. We uploaded two revised manuscript files (pne with and another without tracked changes). We have modified all the points as the referee's suggested

This manuscript provides a review of the use of inhaled corticosteroids in COPD.  The paper is well written and covers the major clinical trials that studied sufficient numbers of subjects to make the results meaningful. The ICS mechanisms section is a nice addition to a review, and helps to explain some of the differences observed in the trials.  Table 2 is also helpful for clinical practice, interpreting study results and trial design.  

General comments:

  • There are over 112 references published using the search ICS and COPD.  Since most of the references cited are older that 2022, I would recommend the authors check for updates.  For example, Chen et al. ( Chest 2023) reviews the impact of ICS on mortality (no I am not one of the authors).

Thanks. We have updated the references and added several newer papers published in 2023. However, we have maintained the total number of references <100 articles

.     For the reader who has a plethora of reviews on the subject to choose from, it might be helpful to state up front why this review is important e.g. new data.  The abstract states that a review of mortality will be discussed, but there is more to this review.

You agree. However, we think that survival is an important outcome (and manufacturers have based their commercial efforts on this topic) and we have  tried to discuss it extensively. In addition, we think that the chapters on side-effects of ICS, difference between ICS molecules and the discussion on some biomarkers can help the interested reader to reflect on the topic that is predated from sponsored works

-I prefer to have the acronym of the study spelled out the first time it is used.

OK; we have changed the paper accordingly

  • It would be helpful to have a section following the review of the trials in the introduction (or somewhere) listing the trial design weaknesses.  For example most are 52 weeks or less, do they account for endotype/phenotype differences in COPD.

This is a very important point. However, there are many reported papers and a new section would change the length of the paper. We have preferred to maintain the current organization of the paper. Possibly, other reviews might discuss this interesting argument

  • Line 64 should have a reference added.

Sorry, but I do not understand this comment. To line 64 I see the search of literature in PubMed

  • There are several references in the literature about % of non-responders to ICS in COPD that would be a nice addition

There are are some references detailing the prevalence of non responders to ICS in COPD, i.e., No 33-35. A comment on the topic was already present: "A prospective 3-year study showed that slightly more than half COPD subjects did not show any exacerbation during the study period and 23% had at least two episodes per year [33]. Other studies have found that approximately one third of COPD subjects may be estimated as common exacerbators [34-35]."

  • Do steroids inhibit neutrophils in COPD?

We have added a comment in 1st paragraph of the Biomarkers section that reports "COPD is characterized by prevalent neutrophilic inflammation, that has poor responsiveness to ICS,......"

-The sections on steroid side effects is well done with respect to risk-benefit.  Perhaps some discussion on reduction of oral steroids by ICS would be of interest.

We have added a comment to the second paragraph of the section 5.3 Systemic side effects "Long-term use of systemic corticosteroids is not recommended in COPD, because it is associated with important systemic adverse events [63]."

  • The effect of patient compliance on utilizing SITTs vs MITTs vs tailoring doses would be an interesting addition.

We have added a comment and some other references on the topic in the first paraagraph of the section 2 History of ICS use in COPD patients"The INvestigation of TRelegy Effectiveness: usual PractIce Design (INTREPID) study showed that the use of SITT improved health status, as evaluated by the COPD Assessment Test (CAT), more than the MITT [5]. Other retrospective or prospective studies including pharmacy data suggest that use of SITT may improve adherence, persistence and, sometimes, symptoms and exacerbations compared to MITT [6-8]."

Round 2

Reviewer 1 Report (Previous Reviewer 1)

Comments and Suggestions for Authors

Good job.

This manuscript is a resubmission of an earlier submission. The following is a list of the peer review reports and author responses from that submission.

Round 1

Reviewer 1 Report

Comments and Suggestions for Authors

Thank you very much for the opportunity to review this publication.

Below you will find some of my suggestions:

Line 14 - I would write: one of the major causes.

Line 37 - According to WHO reports heart diseases still remain nr 1 killer.

Line 40 - beta2- agonists - subscript. 

Line 43 - containing two drugs. 

Line 50 - differences between various types of inhaled glucocorticoids - sounds better.

Line 60-61 - doubled sentence.

 I wonder why you use capital letters in many words e.g. Asthma, Beclomethasone etc. In my opinion, they can be written in lowercase.  

Table 1. Please translate the names of the medicines into English, e.g. vilaterolo, umelidinio (it is umeclidinium).

In my opinion, if the authors want to publish their work in "Biomolecules", they should focus much more on the molecular basis of drugs effectiveness.

Reviewer 2 Report

Comments and Suggestions for Authors

This review by A.S. Melani et. al entitled Inhaled Corticosteroids (ICS) in Subjects with Chronic Obstructive Pulmonary Disease (COPD): an old, unfinished history is indeed aptly named.  Using pub med as a source, since 2020 over 550 publications, including reviews, have been published related to the role of inhaled corticosteroids.  The authors have presented a well referenced compendium of many of the clinical trials initiated to study the appropriate use of ICS in COPD.  A strength of the review is an excellent source of the clinical trials utilizing ICS in COPD.  They have addressed many of the important issues in the debate on the use of ICS in COPD including susceptibility to pneumonia and effects on the microbiome.

General Comments:

1.  My first comment to all the papers I review on COPD, is that COPD is not a single phenotype, specifically emphysema and chronic bronchitis.  Realizing that many COPD subjects often have a mixture, I believe that this should be a consideration for reviews of trials. 

2.  The stated objective of the journal is to publish on molecular mechanisms with biological and medical implications. In section 4 (and throughout the manuscript), the authors describe differences in steroid structure and properties, but could have expanded on this when reviewing the clinical trials and what impact differences between ICS might have had on the outcomes. 

3.  In some sections such as the risk of pneumonia, (section 3.2), the authors present data the argues that ICS are associated with an increase in pneumonia, and then later in the paragraph evidence that suggests the opposite.  The authors conclude with reasons for increased susceptibility, but could have applied to the conclusions of the trials presented.  Similarly, the section on ICS and diabetes presents data associated with, and not associated with.  Are there potential explanations for the discrepancies based on inherent properties of the compounds, or trial design or subjects enrolled?

 4.  The authors use phrases like "seems to be associated" (line 214), "possibly, most physicians feel that.... (line 356) should be referenced.  "the current view of many researchers...(line 401) is there a source for the statement?  

5.  There is a role for pharmacogenetics / genetics in determining appropriate use of ICS in "COPD", especially in defining the phenotypes of this heterogenous disease.  This should deserve some attention in the review.

6.  Since "COPD" is a muco-obstructive disease, there should be some review of the effects of ICS on mucus production and the role in the microbiome.

Minor question:

1.  Line 43- "containing more drugs..."?